# Rapid evolution of the human mutation spectrum

Kelley Harris[1]*, Jonathan K Pritchard[1,2,3]*

[1]Department of Genetics, Stanford University, Stanford, United States; [2]Department of Biology, Stanford University, Stanford, United States; [3]Howard Hughes Medical Institute, Stanford University, Stanford, United States

**Abstract** DNA is a remarkably precise medium for copying and storing biological information. This high fidelity results from the action of hundreds of genes involved in replication, proofreading, and damage repair. Evolutionary theory suggests that in such a system, selection has limited ability to remove genetic variants that change mutation rates by small amounts or in specific sequence contexts. Consistent with this, using SNV variation as a proxy for mutational input, we report here that mutational spectra differ substantially among species, human continental groups and even some closely related populations. Close examination of one signal, an increased TCC→TTC mutation rate in Europeans, indicates a burst of mutations from about 15,000 to 2000 years ago, perhaps due to the appearance, drift, and ultimate elimination of a genetic modifier of mutation rate. Our results suggest that mutation rates can evolve markedly over short evolutionary timescales and suggest the possibility of mapping mutational modifiers.

*For correspondence: kelleyh@ stanford.edu (KH); pritch@ stanford.edu (JKP)

**Competing interests:** The authors declare that no competing interests exist.

## Introduction

Germline mutations not only provide the raw material for evolution but also generate genetic load and inherited disease. Indeed, the vast majority of mutations that affect fitness are deleterious, and hence biological systems have evolved elaborate mechanisms for accurate DNA replication and repair of diverse types of spontaneous damage. Due to the combined action of hundreds of genes, mutation rates are extremely low–in humans, about one point mutation per 100 MB or about 60 genome-wide per generation (*Kong et al., 2012*; *Ségurel et al., 2014*).

While the precise roles of most of the relevant genes have not been fully elucidated, research on somatic mutations in cancer has shown that defects in particular genes can lead to increased mutation rates within very specific sequence contexts (*Alexandrov et al., 2013*; *Helleday et al., 2014*). For example, mutations in the proofreading exonuclease domain of DNA polymerase $\epsilon$ cause TCT→TAT and TCG→TTG mutations on the leading DNA strand (*Shinbrot et al., 2014*). Mutational shifts of this kind have been referred to as 'mutational signatures'. Specific signatures may also be caused by nongenetic factors such as chemical mutagens, UV damage, or guanine oxidation (*Ohno et al., 2014*).

Together, these observations imply a high degree of specialization of individual genes involved in DNA proofreading and repair. While the repair system has evolved to be extremely accurate overall, theory suggests that in such a system, natural selection may have limited ability to fine-tune the efficacy of individual genes (*Lynch, 2011*; *Sung et al., 2012*). If a variant in a repair gene increases or decreases the overall mutation rate by a small amount–for example, only in a very specific sequence context–then the net effect on fitness may fall below the threshold at which natural selection is effective. (Drift tends to dominate selection when the change in fitness is less than the inverse of effective population size). The limits of selection on mutation rate modifiers are especially acute in

**eLife digest** DNA is a molecule that contains the information needed to build an organism. This information is stored as a code made up of four chemicals: adenine (A), guanine (G), cytosine (C), and thymine (T). Every time a cell divides and copies its DNA, it accidentally introduces 'typos' into the code, known as mutations. Most mutations are harmless, but some can cause damage. All cells have ways to proofread DNA, and the more resources are devoted to proofreading, the less mutations occur. Simple organisms such as bacteria use less energy to reduce mutations, because their genomes may tolerate more damage. More complex organisms, from yeast to humans, instead need to proofread their genomes more thoroughly.

Recent research has shown that humans have a lower mutation rate than chimpanzees and gorillas, their closest living relatives. Humans and other apes copy and proofread their DNA with basically the same biological machinery as yeast, which is about a billion years old. Yet, humans and apes have only existed for a small fraction of this time, a few million years. Why then do humans need to replicate and proofread their DNA differently from apes, and could it be that the way mutations arise is still evolving?

Previous research revealed that European people experience more mutations within certain DNA motifs (specifically, the DNA sequences 'TCC', 'TCT', 'CCC' and 'ACC') than Africans or East Asians do.

Now, Harris (who conducted the previous research) and Pritchard have compared how various human ethnic groups accumulate mutations and how these processes differ in different groups.

Statistical analysis of the genomes of thousands of people from all over the world did indeed show that the mutation rates of many different three-letter DNA motifs have changed during the past 20,000 years of human evolution. Harris and Pritchard report that when groups of humans left Africa and settled in isolated populations across different continents, each population quickly became better at avoiding mutations in some genomic contexts, but worse in others. This suggests that the risk of passing on harmful mutations to future generations is changing and evolving at an even faster rate than was originally suspected.

The results suggest that every human ethnic group carries specific variants of the genes which ensure that DNA replication and repair are accurate. These differences appear to influence which types of mutations are frequently passed down to future generations. An important next step will be to identify the genetic variants that could be controlling mutational patterns and how they affect human health.

recombining organisms such as humans because a variant that increases the mutation rate can recombine away from deleterious mutations it generates elsewhere in the genome.

Given these theoretical predictions, we hypothesized that there may be substantial scope for modifiers of mutation rates to segregate within human populations, or between closely related species. Most triplet sequence contexts have mutation rates that vary across the evolutionary tree of mammals (*Hwang and Green, 2004*), but evolution of the mutation spectrum over short time scales has been less well described. Weak natural mutators have recently been observed in yeast (*Bui et al., 2017*) and inferred from human haplotype data (*Seoighe and Scally, 2017*); if such mutators affect specific pathways of proofreading or repair, then we may expect shifts in the abundance of mutations within particular sequence contexts. Indeed, one of us has recently identified a candidate signal of this type, namely an increase in TCC→TTC transitions in Europeans, relative to other populations (*Harris, 2015*); this was recently replicated (*Mathieson and Reich, 2016*). Here, we show that mutation spectrum change is much more widespread than these initial studies suggested: although the TCC→TTC rate increase in Europeans was unusually dramatic, smaller scale changes are so commonplace that almost every great ape species and human continental group has its own distinctive mutational spectrum.

## Results

To investigate the mutational processes in different human populations, we classified each single nucleotide variants (SNV) in the 1000 Genomes Phase 3 data (*Auton et al., 2015*) in terms of its ancestral allele, derived allele, and 5' and 3' flanking nucleotides. We collapsed strand complements together to obtain 96 SNV categories. Since the detection of singletons may vary across samples, and because some singletons may result from cell-line or somatic mutations, we only considered variants seen in more than one copy. We further excluded variants in annotated repeats (since read mapping error rates may be higher in such regions) and in PhyloP conserved regions (to avoid selectively constrained regions) (*Pollard et al., 2010*). From the remaining sites, we calculated the distribution of derived SNVs carried by each Phase 3 individual. We used this as a proxy for the mutational input spectrum in the ancestors of each individual.

To explore global patterns of the mutation spectrum, we performed principal component analysis (PCA) in which each individual was characterized simply by the fraction of their derived alleles in each of the 96 SNV categories (*Figure 1A*). PCA is commonly applied to individual-level genotypes, in which case the PCs are usually highly correlated with geography (*Novembre et al., 2008*). Although the triplet mutation spectrum is an extremely compressed summary statistic compared to typical genotype arrays, we found that it contains sufficient information to reliably classify individuals by continent of origin. The first principal component separated Africans from non-Africans, and the second separated Europeans from East Asians, with South Asians and admixed native Americans (*Figure 1—figure supplement 2*) appearing intermediate between the two.

Remarkably, we found that the mutation spectrum differences among continental groups are composed of small shifts in the abundance of many different mutation types (*Figure 1B*). For example, comparing Africans and Europeans, 43 of the 96 mutation types are significant at a $p<10^{-5}$ threshold using a forward variable selection procedure. The previously described TCC→TTC signature partially drives the difference between Europeans and the other groups, but most other shifts are smaller in magnitude and appear to be spread over more diffuse sets of related mutation types. East Asians have excess A→T transversions in most sequence contexts, as well as about 10% more *AC→*CC mutations than any other group. Compared to Africans, all Eurasians have proportionally fewer C→* mutations relative to A→* mutations.

### Replication of mutation spectrum shifts

One possible concern is that batch effects or other sequencing artifacts might contribute to differences in mutational spectra. Therefore we replicated our analysis using 201 genomes from the Simons Genome Diversity Project (*Mallick et al., 2016*). The SGDP genomes were sequenced at high coverage, independently from 1000 Genomes, using an almost non-overlapping panel of samples. We found extremely strong agreement between the mutational shifts in the two data sets (*Figure 2*). For example, all of the 43 mutation types with a significant difference between Africa and Europe (at $p<10^{-5}$) in 1000 Genomes also show a frequency difference in the same direction in SGDP (comparing Africa and West Eurasia). In both 1000 Genomes and SGDP, the enrichment of *AC→*CC mutations in East Asia is larger in magnitude than any other signal aside from the previously described TCC→TTC imbalance.

The greatest discrepancies between 1000 Genomes and SGDP involve transversions at CpG sites, which are among the rarest mutational classes. These discrepancies might result from data processing differences or random sampling variation, but might also reflect differences in the fine-scale ethnic composition of the two panels.

### Evidence for a pulse of TCC→TTC mutations in Europe and South Asia

To investigate the timescale over which the mutation spectrum change occurred, we analyzed the allele frequency distribution of TCC→TTC mutations, which are highly enriched in Europeans (*Figure 3A*; $p<1 \times 10^{-300}$ for Europe vs. Africa) and to a lesser extent in South Asians. We calculated allele frequencies both in 1000 Genomes and in the larger UK10K genome panel (*Walter et al., 2015*). As expected for a signal that is primarily European, we found particular enrichment of these mutations at low frequencies. But surprisingly, the enrichment peaks around 0.6% frequency in UK10K, and there is practically no enrichment among the very lowest frequency variants (*Figure 3B* and *Figure 3—figure supplement 1*). C→T mutations on other backgrounds, namely within TCT,

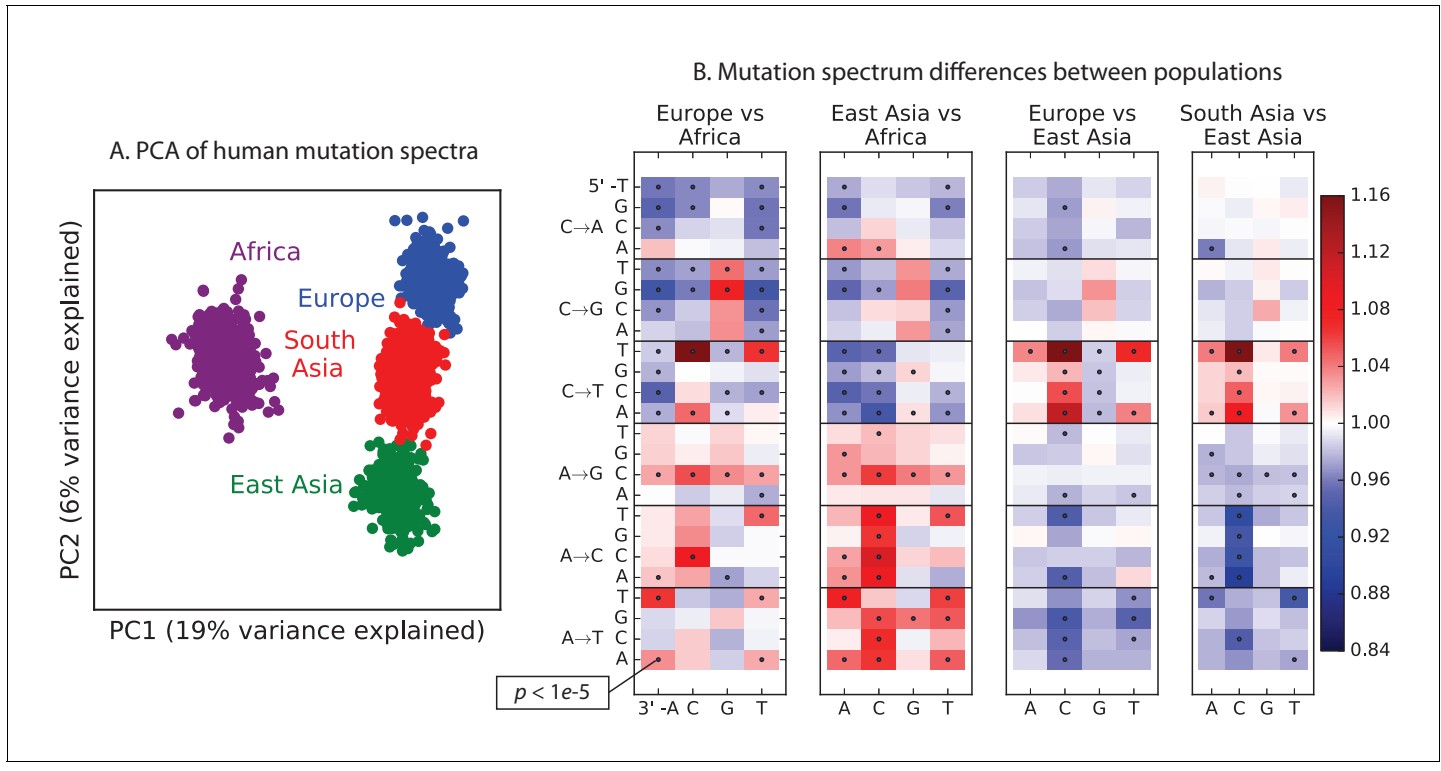

**Figure 1.** Global patterns of variation in SNV spectra. (A) Principal component analysis of individuals according to the fraction of derived alleles that each individual carries in each of 96 mutational types. (B) Heatmaps showing, for pairs of continental groups, the ratio of the proportions of SNVs in each of the 96 mutational types. Each block corresponds to one mutation type; within blocks, rows indicate the 5' nucleotide, and columns indicate the 3' nucleotide. Red colors indicate a greater fraction of a given mutation type in the first-listed group relative to the second. Points indicate significant contrasts at $p < 10^{-5}$. See *Figure 1—figure supplements 1*, *2* and *3* for heatmap comparisons between additional population pairs as well as a description of PCA loadings and the p-valuesof all mutation class enrichments. *Figure 1—figure supplement 4* demonstrates that these patterns are unlikely to be driven by biased gene conversion. In *Figure 1—figure supplement 5*, we see that this mutation spectrum structure replicates on both strands of the transcribed genome as well as the non-transcribed portion of the genome. *Figure 1—figure supplements 6*, *7* and *8* show that most of this structure replicates across multiple chromatin states and varies little with replication timing.

The following source data and figure supplements are available for figure 1:

**Source data 1.** This text file shows the number of SNPs in each of the 96 mutational categories that passed all filters in each 1000 Genomes continental group.

**Figure supplement 1.** Pairwise mutation spectrum comparisons among continental groups.

**Figure supplement 2.** PCA of all 1000 Genomes continental groups.

**Figure supplement 3.** Mutation spectrum comparison p-values.

**Figure supplement 4.** The effects of biased gene conversion on mutation spectra.

**Figure supplement 5.** Mutation spectra of transcribed vs non-transcribed DNA.

**Figure supplement 6.** Mutation spectra of ChromHMM chromatin states (Part I of II).

**Figure supplement 7.** Mutation spectra of ChromHMM chromatin states (Part II of II).

**Figure supplement 8.** Variation of the mutation spectrum with DNA replication timing.

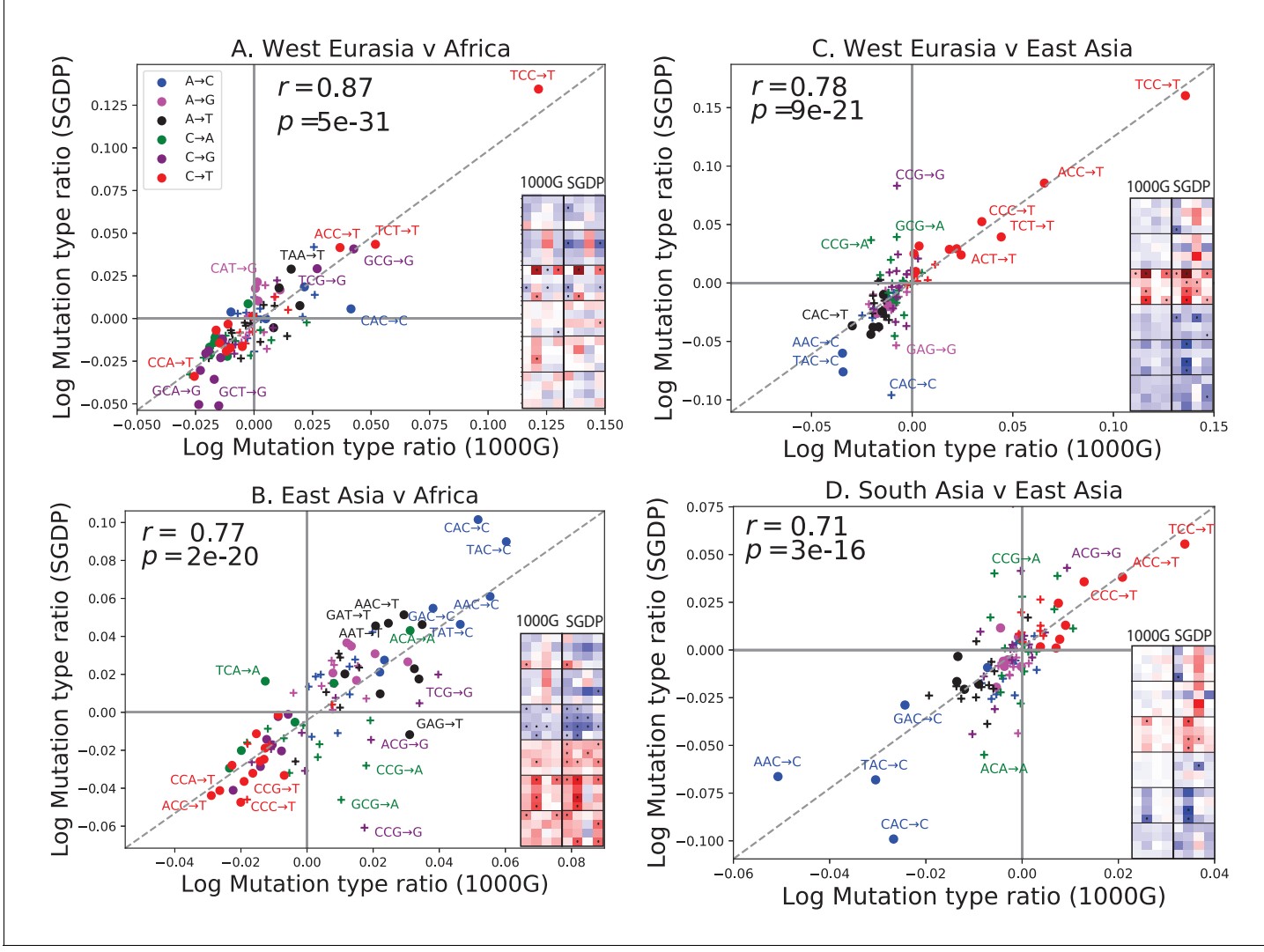

**Figure 2.** Concordance of mutational shifts in 1000 Genomes versus SGDP. Each panel shows natural-log mutation spectrum ratios between a pair of continental groups, based on 1000 Genomes (x-axis) and SGDP (y-axis) data. Data points encoded by (+) symbols denote mutation types that are not significantly enriched in either population in the *Figure 1* 1000 Genomes analysis (*p*<10⁻⁵). These heatmaps use the same labeling and color scale as in *Figure 1*. All 1000 Genomes ratios in this figure were estimated after projecting the 1000 Genomes site frequency spectrum down to the sample size of SGDP. See *Figure 2—figure supplements 1* and *2* for a complete set of SGDP heatmaps and regressions versus 1000 Genomes.

The following figure supplements are available for figure 2:

**Figure supplement 1.** Heatmap comparisons between continental groups in 1000 Genomes and the SGDP.

**Figure supplement 2.** Regression of the SGDP heatmap coefficients versus the corresponding 1000 Genomes heatmap coefficients.

CCC and ACC contexts, are also enriched in Europe and South Asia and show a similar enrichment around 0.6% frequency that declines among rarer variants (*Figure 3C*). This suggests that these four mutation types comprise the signature of a single mutational pulse that is no longer active. No other mutation types show such a pulse-like distribution in UK10K, although several types show evidence of monotonic rate change over time (*Figure 3—figure supplements 3*, *4* and *5*).

We used the enrichment of TCC→TTC mutations as a function of allele frequency to estimate when this mutation pulse was active. Assuming a simple piecewise-constant model, we infer that the rate of TCC→TTC mutations increased dramatically ~15,000 years ago and decreased again ~2000 years ago. This time-range is consistent with results showing this signal in a pair of prehistoric

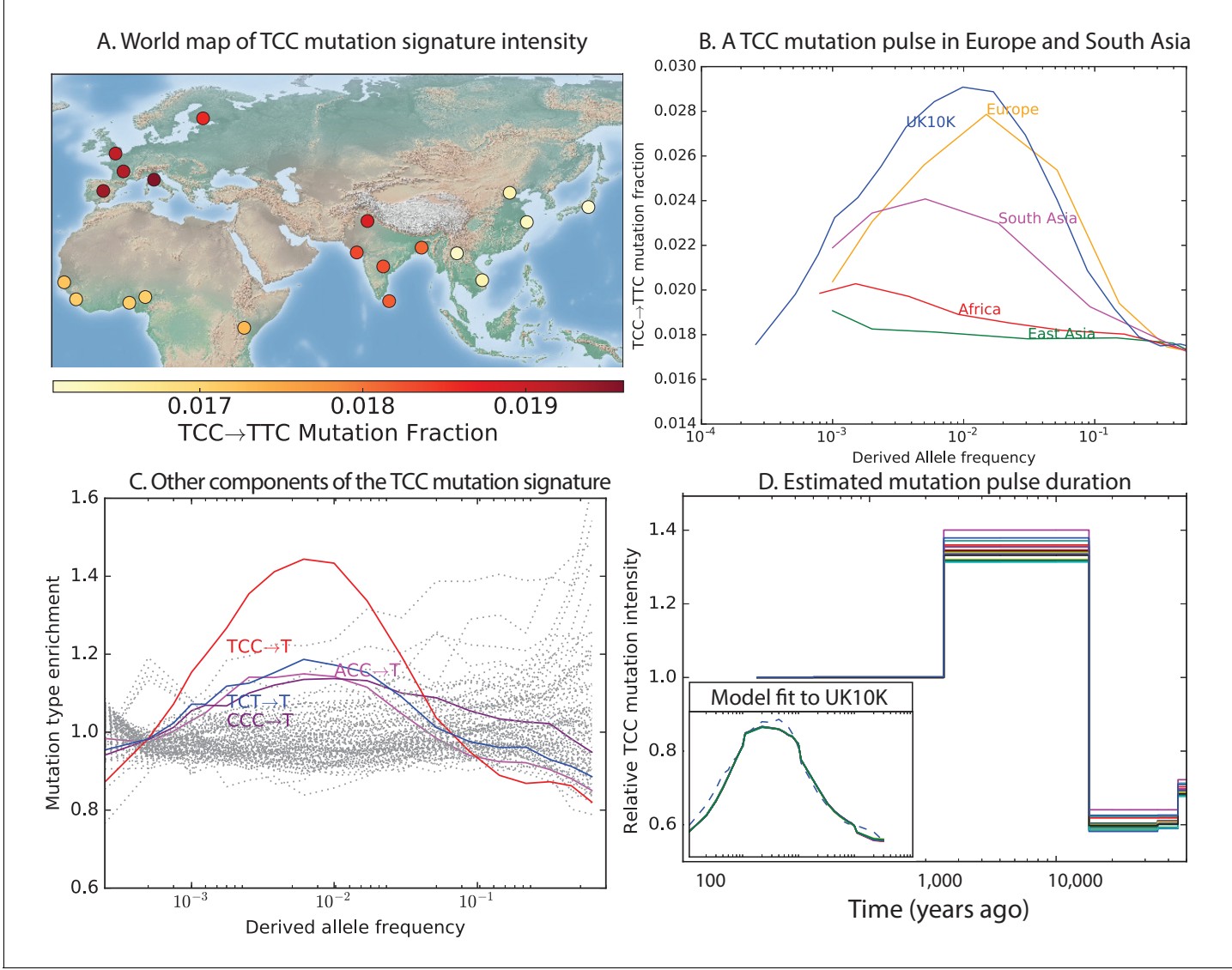

**Figure 3.** Geographic distribution and age of the TCC mutation pulse. (**A**) Observed frequencies of TCC→TTC variants in 1000 Genomes populations. (**B**) Fraction of TCC→TTC variants as a function of allele frequency in different samples indicates that these peak around 1%. See *Figure 3—figure supplement 1* for distributions of TCC→TTC allele frequency within all 1000 Genomes populations, and see *Figure 3—figure supplement 2* for the replication of this result in the Exome Aggregation Consortium Data. In the UK10K data, which has the largest sample size, the peak occurs at 0.6% allele frequency. (**C**) Other enriched C→T mutations with similar context also peak at 0.6% frequency in UK10K. See *Figure 3—figure supplements 3, 4* and *5* for labeled allele frequency distributions of all 96 mutation types (most represented here as unlabeled grey lines). See *Figure 3—figure supplement 6* for heatmap comparisons of the 1000 Genomes populations partitioned by allele frequency, which provide a different view of these patterns. (**D**) A population genetic model supports a pulse of TCC→TTC mutations from 15,000 to 2000 years ago. Inset shows the observed and predicted frequency distributions of this mutation under the inferred model.

The following figure supplements are available for figure 3:

**Figure supplement 1.** TCC→TTC mutation fraction as a function of allele frequency in all 1000 Genomes populations.

**Figure supplement 2.** Fraction of TCC→TTC mutations as a function of allele frequency in ExAC.

**Figure supplement 3.** Mutation type enrichment as a function of allele frequency in UK10K (Part I of III).

**Figure supplement 4.** Mutation type enrichment as a function of allele frequency in UK10K (Part II of III).

*Figure 3 continued on next page*

*Figure 3 continued*

**Figure supplement 5.** Mutation type enrichment as a function of allele frequency in UK10K (Part III of III).

**Figure supplement 6.** Mutation spectrum comparisons partitioned by allele frequency.

European samples from 7000 and 8000 years ago, respectively (*Mathieson and Reich, 2016*). We hypothesize that this mutation pulse may have been caused by a mutator allele that drifted up in frequency starting 15,000 years ago, but that is now rare or absent from present day populations.

Although low frequency allele calls often contain a higher proportion of base calling errors than higher frequency allele calls do, it is not plausible that base-calling errors could be responsible for the pulse we have described. In the UK10K data, a minor allele present at 0.6% frequency corresponds to a derived allele that is present in 23 out of 3854 sampled haplotypes and supported by 80 short reads on average (assuming 7x coverage per individual). When independently generated datasets of different sizes are projected down to the same sample size, the TCC→TTC pulse spans the same range of allele frequencies in both datasets (*Figure 3—figure supplements 1* and *2*), which would not be the case if the shape of the curve were a function of low-frequency errors.

## Fine-scale mutation spectrum variation in other populations

Encouraged by these results, we sought to find other signatures of recent mutation pulses. We generated heatmaps and PCA plots of mutation spectrum variation within each continental group, looking for fine-scale differences between closely related populations (*Figure 4* and *Figure 4—figure supplement 1* through 6). In some cases, mutational spectra differ even between very closely related populations. For example, the *AC→*CC mutations with elevated rates in East Asia appear to be distributed heterogeneously within that group, with most of the load carried by a subset of the Japanese individuals. These individuals also have elevated rates of ACA→AAA and TAT→TTT mutations (*Figure 4A* and *Figure 4—figure supplement 4*). This signature appears to be present in only a handful of Chinese individuals, and no Kinh or Dai individuals. As seen for the European TCC mutation, the enrichment of these mutation types peaks at low frequencies, that is, ~1%. Given the availability of only 200 Japanese individuals in 1000 Genomes, it is hard to say whether the true peak is at a frequency much lower than 1%.

PCA reveals relatively little fine-scale structure within the mutational spectra of Europeans or South Asians (*Figure 4—figure supplements 5* and *6*). However, Africans exhibit some substructure (*Figure 4—figure supplement 3*), with the Luhya exhibiting the most distinctive mutational spectrum. Unexpectedly, a closer examination of PC loadings reveals that the Luhya outliers are enriched for the same mutational signature identified in the Japanese. Even in Europeans and South Asians, the first PC is heavily weighted toward *AC→*CC, ACA→AAA, and TAT→TTT, although this signature explains less of the mutation spectrum variance within these more homogeneous populations. The sharing of this signature may suggest either parallel increases of a shared mutation modifier, or a shared aspect of environment or life history that affects the mutation spectrum.

## Mutation spectrum variation among the great apes

Finally, given our finding of extensive fine-scale variation in mutational spectra between human populations, we hypothesized that mutational variation between species is likely to be even greater. To compare the mutation spectra of the great apes in more detail, we obtained SNV data from the Great Ape Diversity Panel, which includes 78 whole genome sequences from six great ape species including human (*Prado-Martinez et al., 2013*). Overall, we find dramatic variation in mutational spectra among the great ape species (*Figure 5* and *Figure 5—figure supplement 1*).

As noted previously (*Moorjani et al., 2016a*), one major trend is a higher proportion of CpG mutations among the species closest to human, possibly reflecting lengthening generation time along the human lineage, consistent with previous indications that species closely related to humans have lower mutation rates than more distant species (*Goodman, 1961*; *Li and Tanimura, 1987*; *Scally and Durbin, 2012*). However, most other differences are not obviously related to known processes such as biased gene conversion and generation time change. The A→T mutation rate appears

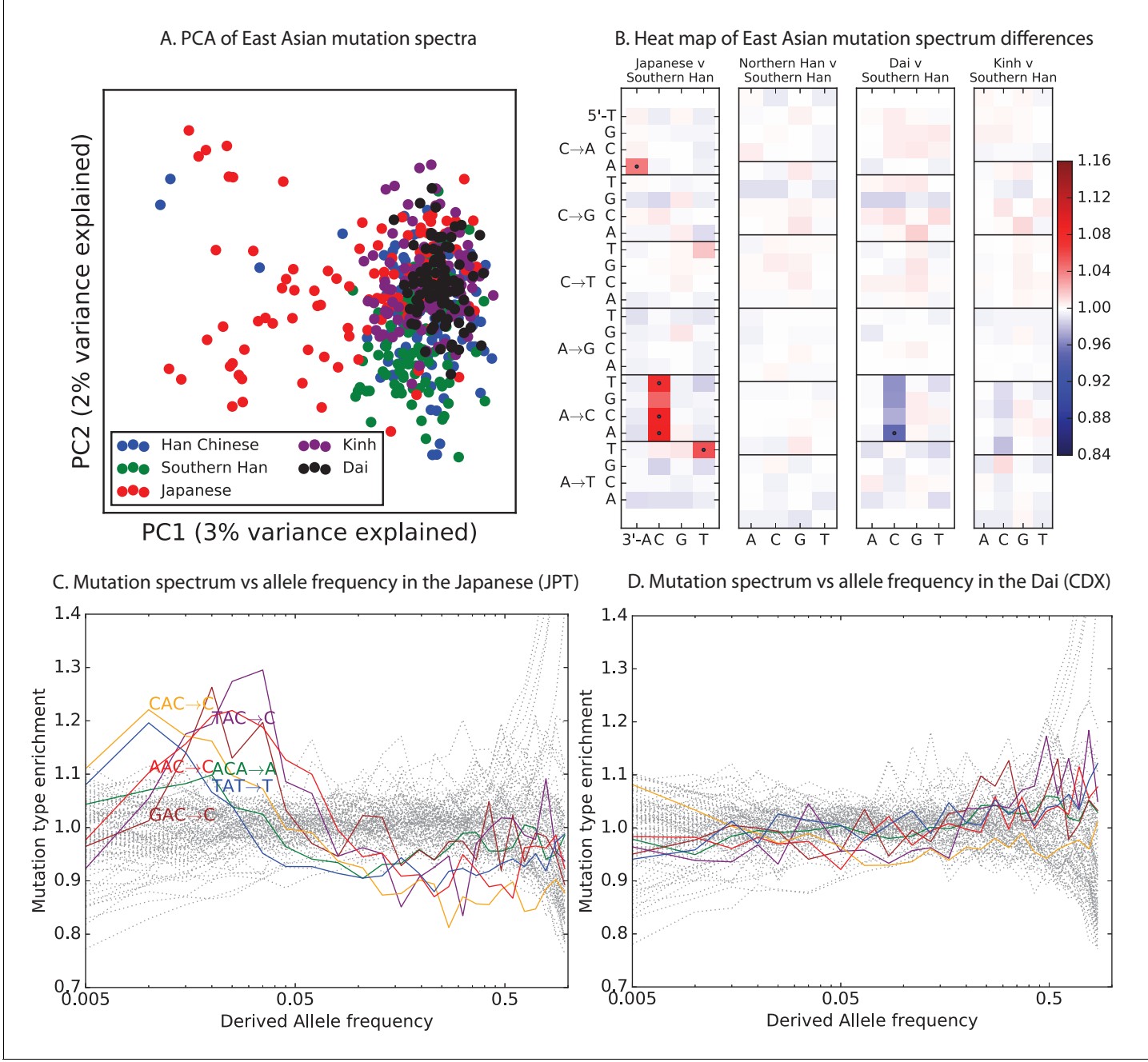

**Figure 4.** Mutational variation among east Asian populations. (**A**) PCA of east Asian samples from 1000 Genomes, based on the relative proportions of each of the 96 mutational types. See *Figure 4—figure supplement 2* through 6 for other finescale population PCAs. (**B**) Heatmaps showing, for pairs of east Asian samples, the ratio of the proportions of SNVs in each of the 96 mutational types. Points indicate significant contrasts at $p < 10^{-5}$. See *Figure 4—figure supplement 1* for additional finescale heatmaps. (**C**) and (**D**) Relative enrichment of each mutational type in Japanese and Dai, respectively as a function of allele frequency. Six mutation types that are enriched in JPT are indicated. Populations: CDX=Dai, CHB=Han (Beijing); CHS=Han (south China); KHV=Kinh; JPT=Japanese.

The following source data and figure supplements are available for figure 4:

**Source data 1.** This text file shows the number of SNPs in each of the 96 mutational categories that passed all filters in each finescale 1000 Genomes population.

**Figure supplement 1.** Mutation spectrum differences within Africa, Europe, East Asia, and South Asia.

*Figure 4 continued on next page*

**Figure supplement 2.** PCA of American populations.
DOI: 10.7554/eLife.24284.026
**Figure supplement 3.** PCA of African populations.
DOI: 10.7554/eLife.24284.027
**Figure supplement 4.** PCA of East Asian populations.
DOI: 10.7554/eLife.24284.028
**Figure supplement 5.** PCA of South Asian populations.
DOI: 10.7554/eLife.24284.029
**Figure supplement 6.** PCA of European populations.
DOI: 10.7554/eLife.24284.030

to have sped up in the common ancestor of humans, chimpanzees, and bonobos, a change that appears consistent with a mutator variant that was fixed instead of lost. It is unclear whether this ancient A→T speedup is related to the A→T speedup in East Asians. Other mutational signatures appear on only a single branch of the great ape tree, such as a slowdown of A→C mutations in gorillas.

## Discussion

The widespread differences captured in *Figures 1* and *2* may be footprints of allele frequency shifts affecting different mutator alleles. But in principle, other genetic and non-genetic processes may also impact the observed mutational spectrum. First, biased gene conversion (BGC) tends to favor C/G alleles over A/T, and BGC is potentially more efficient in populations of large effective size compared to populations of smaller effective size (*Galtier et al., 2001*). However, despite the

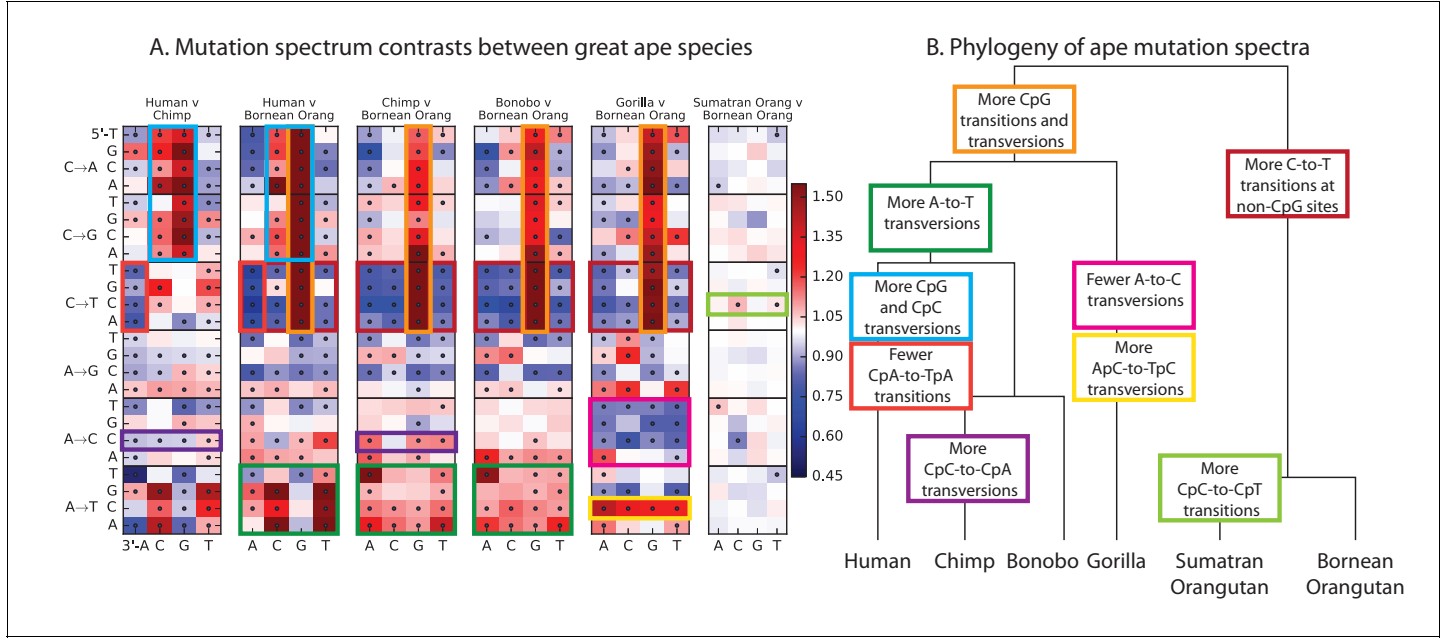

**Figure 5.** Mutational differences among the great apes. (**A**) Relative abundance of SNV types in 5 ape species compared to Bornean Orangutan; data from (*Prado-Martinez et al., 2013*). Boxes indicate labels in (**B**). For additional comparisons see *Figure 5—figure supplement 1*. (**B**) Schematic phylogeny of the great apes highlighting notable changes in SNV abundance.

The following figure supplement is available for figure 5:

**Figure supplement 1.** Mutation spectra of great apes.

bottlenecks that are known to have affected Eurasian diversity, there is no clear trend of an increased fraction of C/G→A/T relative to A/T→C/G in non-Africans vs Africans, or with distance from Africa (*Figure 1—figure supplement 7*), and previous studies have also found little evidence for a strong genome-wide effect of BGC on the mutational spectrum in humans and great apes (*Do et al., 2015*; *Moorjani et al., 2016a*). For these reasons, we think that evolution of the mutational process is a better explanation than BGC or selection for differences that have been observed between the spectra of ultra-rare singleton variants and older human genetic variation (*Carlson et al., 2017*);

It is also known that shifts in generation time or other life-history traits may affect mutational spectra, particularly for CpG transitions (*Martin and Palumbi, 1993*; *Amster and Sella, 2016*). Most CpG transitions result from spontaneous methyl-cytosine deamination as opposed to errors in DNA replication. Hence the rate of CpG transitions is less affected by generation time than other mutations (*Hwang and Green, 2004*; *Moorjani et al., 2016b*; *Gao et al., 2016*). We observe that Europeans have a lower fraction of CpG variants compared to Africans, East Asians and South Asians (*Figure 1B*), consistent with a recent report of a lower rate of de novo CCG→CTG mutations in European individuals compared to Pakistanis (*Narasimhan et al., 2016*). Such a pattern may be consistent with a shorter average generation time in Europeans (*Moorjani et al., 2016b*), although it is unclear that a plausible shift in generation-time could produce such a large effect. Apart from this, the other patterns evident in *Figure 1* do not seem explainable by known processes.

In summary, we report here that, mutational spectra differ significantly among closely related human populations, and that they differ greatly among the great ape species. Our work shows that subtle, concerted shifts in the frequencies of many different mutation types are more widespread than dramatic jumps in the rate of single mutation types, although the existence of the European TCC→TTC pulse shows that both modes of evolution do occur (*Harris, 2015*; *Moorjani et al., 2016b*; *Mathieson and Reich, 2016*).

At this time, we cannot exclude a role for nongenetic factors such as changes in life history or mutagen exposure in driving these signals. However, given the sheer diversity of the effects reported here, it seems parsimonious to us to propose that most of this variation is driven by the appearance and drift of genetic modifiers of mutation rate. This situation is perhaps reminiscent of the earlier observation that genome-wide recombination patterns are variable among individuals (*Coop et al., 2008*), and ultimate discovery of PRDM9 (*Baudat et al., 2010*); although in this case it is unlikely that a single gene is responsible for all signals seen here. As large datasets of de novo mutations become available, it should be possible to map mutator loci genome-wide. In summary, our results suggest the likelihood that mutational modifiers are an important part of the landscape of human genetic variation.

# Materials and methods

## Data availability
All datasets analyzed here are publicly available at the following websites:

| | |
|---|---|
| 1000 Genomes phase 3 | http://www.1000genomes.org/category/phase-3/ |
| UK10K | http://www.uk10k.org/data-access.html |
| Simons Genome Diversity Panel | https://www.simonsfoundation.org/life-sciences/simons-genome-diversity-project-dataset/ |

## Human mutation spectrum processing
Mutation spectra were computed using 1000 Genomes Phase 3 SNPs (*Auton et al., 2015*) that are biallelic, pass all 1000 Genomes quality filters, and are not adjacent to any N's in the hg19 reference sequence. Ancestral states were assigned using the UCSC Genome Browser alignment of hg19 to the PanTro2 chimpanzee reference genome; SNPs were discarded if neither the reference nor alternate allele matched the chimpanzee reference. To minimize the potential impact of ancestral

misidentification errors, SNPs with derived allele frequency higher than 0.98 were discarded. We also filtered out regions annotated as 'conserved' based on the 100-way PhyloP conservation score (*Pollard et al., 2010*), download from http://hgdownload.cse.ucsc.edu/goldenPath/hg19/phast-Cons100way/, as well as regions annotated as repeats by RepeatMasker (*Smit et al., 2013*), downloaded from http://hgdownload.cse.ucsc.edu/goldenpath/hg19/database/nestedRepeats.txt.gz. To be counted as part of the mutation spectrum of population *P* (which can be either a continental group or a finer-scale population from one city), a SNP should not be a singleton within population *P*–at least two copies of the ancestral and derived alleles must be present within that population.

An identical approach was used to extract the mutation spectrum of the UK10K ALSPAC panel (*Walter et al., 2015*), which is not subdivided into smaller populations. The data were filtered as described in *Field et al. (2016)*. The filtering procedure performed by *Field et al. (2016)* reduces the ALSPAC sample size to 1927 individuals.

We also computed mutation spectra of the Simons Genome Diversity Panel (SGDP) populations (*Mallick et al., 2016*). Four of the SGDP populations, West Eurasia, East Asia, South Asia, and Africa, were compared to their direct counterparts in the 1000 Genomes data. Three additional SGDP populations, Central Asia and Siberia, Oceania, and America, had no close 1000 Genomes counterparts and were not analyzed here (although each project contained a panel of people from the Americas, the composition of the American panels was extremely different, with the 1000 Genomes populations being much more admixed with Europeans and Africans). SGDP sites with more than 20% missing data were not utilized. All other data processing was done the same way described for the 1000 Genomes data.

The following table gives the same size of each population panel, as well as the total number of SNPs segregating in the panel that are used to compute mutation type ratios:

| Dataset | Population | Number of individuals | Number of SNPs |
| --- | --- | --- | --- |
| 1 kg | Africa | 504 | 16,870,400 |
| 1 kg | Europe | 503 | 8,508,040 |
| 1 kg | East Asia | 504 | 7,895,925 |
| 1 kg | South Asia | 489 | 9,552,781 |
| SGDP | Africa | 45 | 6,569,658 |
| SGDP | West Eurasia | 69 | 4,201,571 |
| SGDP | East Asia | 49 | 3,312,645 |
| SGDP | South Asia | 38 | 3,449,624 |

## Great ape diversity panel data processing

Biallelic great ape SNPs were extracted from the Great Ape Diversity Panel VCF (*Prado-Martinez et al., 2013*), which is aligned to the hg18 human reference sequence. Ancestral states were assigned using the Great Ape Genetic Diversity project annotation, which used the Felsenstein pruning algorithm to assign allelic states to internal nodes in the great ape tree. In the Great Ape Diversity Panel, the most recent common ancestor (MRCA) of the human species is labeled as node 18; the MRCAs of chimpanzees, bonobos, gorillas, and orangutans, respectively, are labeled as node 16, node 17, node 19, and node 15. We extracted the state of each MRCA at each SNP in the alignment and used it to polarize the ancestral and derived allele at that site; a SNP was discarded whenever the ancestral node was assigned an uncertain or polymorphic ancestral state. As with the human data, SNPs with derived allele frequency higher than 0.98 were not used, and both repeats and PhyloP-annotated conserved regions were filtered away.

## Visual representation of mutation spectra

The mutation type of an SNP is defined in terms of its ancestral allele, its derived allele, and its two immediate 5' and 3' neighbors. Two mutation types are considered equivalent if they are strand-

complementary to each other (e.g. ACG→ATG is equivalent to CGT→CAT). This scheme classifies SNPs into 96 different mutation types, each that can be represented with an A or C ancestral allele.

To compute the frequency $f_P(m)$ of SNP $m$ in population $P$, we count up all SNPs of type $m$ where the derived allele is present in at least one representative of population $P$ (which can be either a specific population such as YRI or a broader continental group such as AFR). After obtaining this count $C_P(m)$, we define $f_P(m)$ to be the ratio $C_P(m)/\sum_{m'} C_P(m')$, where the sum in the denominator ranges over all 96 mutation types $m'$. The enrichment of mutation type $m$ in population $P_1$ relative to population $P_2$ is defined to be $f_{P_1}(m)/f_{P_2}(m)$; these enrichments are visualized as heat maps in *Figures 1B*, *3B* and *4A*.

To track changes in the mutational spectrum over time, we compute $f_P(m)$ in bins of restricted allele frequency. This involves counting the number of SNPs of type $m$ that are present at frequency $\phi$ in population $P$ to obtain counts $C_P(m, \phi)$ and frequencies $f_P(m, \phi) = C_P(m, \phi)/\sum_{m'} C_P(m'\phi)$. Deviation of the ratio $f_P(m, \phi)/f_P(m)$ from one indicates that the rate of $m$ has fluctuated recently in the history of population $P$. To make the sampling noise approximately uniform across alleles of different frequencies, alleles of derived count greater than five were grouped into approximately log-spaced bins that each contained similar numbers of UK10K SNPs. More precisely, we defined a set of bin endpoints $b_1, b_2, \ldots$ such that the total number of SNPs ranging in derived allele count between $b_i$ and $b_{i+1} - 1$ is greater than or equal to the number of 5-ton SNPs, while the total number of SNPs ranging in derived allele count from $b_i$ to $b_{i+1} - 2$ is less than the number of 5-ton SNPs.

In some cases, for example *Figure 2*, *Figure 2—figure supplement 1B* and *Figure 3—figure supplement 1*, site frequency spectra were projected down to a smaller sample size before counting SNPs in order to more accurately compare datasets of different sample sizes. A binomial sampling approach was used to project a sample of $N$ haplotypes does to a smaller sample size $n$. Letting $C_P^{(N)}(m, \phi)$ denote the SNP counts in the large sample of $N$ haplotypes, effective SNP counts $C_P^{(n)}(m, \phi)$ in a sample of $n$ haplotypes are computed as follows:

$$C_P^{(n)}(m, k/n) = \binom{n}{k} \sum_{\ell=1}^{N-1} (\ell/N)^k (1 - \ell/N)^{n-k} C_P^{(N)}(m, \ell/N)$$

## Significance testing

One central goal of this paper is to test whether many mutation types differ in rate between human populations or whether mutation spectrum shifts have been rare events affecting only a small proportion of mutation types. A simple statistical method for answering this question would be to perform 96 separate chi-square tests, one for each triplet-context-dependent mutation type, as follows:

Let $S_i$ denote the total number of SNPs segregating in population $P_i$, and let $S_i^{(m)}$ denote the number of SNPs of mutation type $m$. If mutation type $m$ is more prevalent in population $P_1$ than in population $P_2$, a chi-square test provides a natural way of assessing the significance of this difference. As described in *Harris (2015)*, this test is performed on the following two-by-two contingency table:

| $S_1^{(m)}$ | $P_1 - S_1^{(m)}$ |
|---|---|
| $S_2^{(m)}$ | $P_2 - S_2^{(m)}$ |

It would be appealing to conclude that every mutation type 'passing' this chi-square test is a mutation type that has changed in rate during recent human history. However, if we were to perform the full set of 96 tests, they would not be independent. A sufficiently large increase in the rate of one mutation type $m_1$ in population $P_1$ after divergence from $P_2$ could cause another mutation type $m_2$, whose rate has remained constant, to comprise significantly different fractions of the SNPs from $P_1$ and $P_2$. To minimize this effect, we formulate the following iterative procedure of conditionally independent tests: first, compute a chi-square significance value $p_{\text{unordered}}(m)$ for each mutation type $m$ using the two-by-two chi-square table above. We then use these values to order the SNPs from lowest p value to highest and compute a set of ordered p values $p_{\text{ordered}}(m)$. For the mutation type $m_0$ with the lowest unordered p value, $p_{\text{unordered}}(m_0) = p_{\text{ordered}}(m_0)$. For mutation type $m_i$, which has

the $i$th lowest unordered $p$ value and $i<96$, $p_{\text{ordered}}(m_i)$ is computed from the following contingency table:

$$
\begin{array}{cc}
S_1^{(m_i)} & \sum_{j=i+1}^{96} S_1^{(m_j)} \\
S_2^{(m_i)} & \sum_{j=i+1}^{96} S_2^{(m_j)}
\end{array}
$$

For mutation type $m_{96}$, which has the highest unordered $p$ value, the ordered $p$ value is computed from the contingency table

$$
\begin{array}{cc}
S_1^{(m_{96})} & S_1^{(m_{95})} \\
S_2^{(m_{96})} & S_2^{(m_{95})}
\end{array}
$$

This procedure is guaranteed to find fewer mutation types to differ significantly in rate between populations compared to separate chi-square tests.

## Principal component analysis

The python package matplotlib.mlab.PCA was used to perform PCA on the complete set of 1000 Genomes diploid genomes. First, the triplet mutational spectrum of each haplotype $h$ was computed as a 96-element vector encoding the mutation frequencies $(f_h(m))_m$ of the non-singleton derived alleles present on that haplotype. The mutational spectrum of each diploid genome was then computed by averaging together the spectra of its two constituent haplotypes. In the same way, a separate PCA was performed on each of the five continental groups to reveal finescale components of mutation spectrum variation.

## Dating of the TCC→T mutation pulse

We estimated the duration and intensity of TCC→T rate acceleration in Europe by fitting a simple piecewise-constant rate model to the UK10K frequency data. To specify the parameters of the model, we divide time into discrete log-spaced intervals bounded by time points $t_1, ..., t_d$, assigning each interval a TCC→T mutation rate $r_0, ...r_d$. In units of generations before the present, the time discretization points were chosen to be: 20, 40, 200, 400, 800, 1200, 1600, 2000, 2400, 2800, 3200, 3600, 4000, 8000, 12,000, 16,000, 20,000, 24,000, 28,000, 32,000, 36,000, 40,000. We assume that the total rate $r$ of mutations other than TCC→T stays constant over time (a first-order approximation).

In terms of these rate variables, we can calculate the expected shape of the TCC→T pulse shown in *Figure 2B* of the main text. The shape of this curve depends on both the mutation rate parameters $r_i$ and the demographic history of the European population, which determines the joint distribution of allele frequency and allele age. To account for the effects of demography, we use Hudson's ms program to simulate 10,000 random coalescent trees under a realistic European demographic history inferred from allele frequency data (*Tennessen et al., 2012*) and condition our inference upon this collection of trees as follows:

Let $A(m, t)$ be the function for which $\int_{t_i}^{t_{i+1}} A(m, t)dt$ equals the coalescent tree branch length, averaged over the sample of simulated trees, that is ancestral to exactly $m$ lineages and falls between time $t_i$ and $t_{i+1}$. Given this function, which can be empirically estimated from a sample of simulated trees, the expected frequency spectrum entry $k/n$ is

$$
E(k/n) = \frac{\sum_{i=1}^{d} \int_{t_{i-1}}^{t_i} A(k,t)dt}{\sum_{j=1}^{n} \sum_{i=1}^{d} \int_{t_{i-1}}^{t_i} A(j,t)dt}
$$

and the expected fraction of TCC→T mutations in allele frequency bin $k/n$ is

$$
E(f_{\text{TCC}\to\text{T}}(k/n)) = \frac{\sum_{i=1}^{d} r_i \int_{t_{i-1}}^{t_i} A(k,t)dt}{r \sum_{i=1}^{d} \int_{t_{i-1}}^{t_i} A(k,t)dt}.
$$

The expected value of the TCC→T enrichment ratio being plotted in *Figure 2B* is

$$E(r_{\text{TCC}\to\text{T}}(k/n)) = \frac{\sum_{i=1}^{d} r_i \int_{t_{i-1}}^{t_i} A(k,t)dt \cdot \sum_{j=1}^{n} \sum_{i=1}^{d} \int_{t_{i-1}}^{t_i} A(j,t)dt}{\sum_{i=1}^{d} \int_{t_{i-1}}^{t_i} A(k,t)dt \cdot \sum_{j=1}^{n} \sum_{i=1}^{d} r_i \int_{t_{i-1}}^{t_i} A(j,t)dt}$$

In *Figure 2B*, enrichment ratios are not computed for every allele frequency in isolation, but for allele frequency bins that each contain similar numbers of SNPs. Given integers $1 \leq k_m < k_{m+1} \leq n$, the expected TCC→T enrichment ratio averaged over all SNPs with allele frequency between $k_m/n$ and $k_{m+1}/n$ is:

$$E(r_{\text{TCC}\to\text{T}}(k_m/n)) = \frac{\sum_{i=1}^{d} r_i \int_{t_{i-1}}^{t_i} \sum_{k=k_m}^{k_{m+1}} A(k,t)dt \cdot \sum_{j=1}^{n} \sum_{i=1}^{d} \int_{t_{i-1}}^{t_i} A(j,t)dt}{\sum_{i=1}^{d} \int_{t_{i-1}}^{t_i} \sum_{k=k_m}^{k_{m+1}} A(k,t)dt \cdot \sum_{j=1}^{n} \sum_{i=1}^{d} r_i \int_{t_{i-1}}^{t_i} A(j,t)dt}$$

We optimize the mutation rates $r_1,\ldots,r_d$ using a log-spaced quantization of allele frequencies $k_1/n,\ldots,k_m/n$ defined such that all bins contain similar numbers of SNPs. The chosen allele count endpoints $k_1,\ldots,k_m$ are: 1, 2, 3, 4, 5, 6, 7, 8, 9, 10, 20, 30, 40, 50, 60, 70, 80, 90, 100, 200, 300, 400, 500, 600, 700, 800, 900, 1000, 2000, 3000, 4000. Given this quantization of allele frequencies, we optimize $r_1,\ldots,r_d$ by using the BFGS algorithm to minimize the least squares distance $D(r_0,\ldots,r_d)$ between $E(r_{\text{TCC}\to\text{T}}(k_m/n))$ and the empirical ratio $r_{\text{TCC}\to\text{T}}(k_m/n)$ computed from the UK10K data. This optimization is subject to a regularization penalty that minimizes the jumps between adjacent mutation rates $r_i$ and $r_{i+1}$:

$$D(r_0,\ldots,r_d) = \sum_{m=1}^{d}(E(r_{\text{TCC}\to\text{T}}(k_m/n)) - r_{\text{TCC}\to\text{T}}(k_m/n))^2 + 0.25\sqrt{\sum_{i=1}^{d}(r_{i-1}-r_i)^2}$$

Although the underlying model of mutation rate change assumed here is very simple, it still represents an advance over the method used in (*Harris, 2015*) to estimate of the timing of the TCC→TTC mutation rate increase. That method relied upon explicit estimates of allele age from a dataset of less than 100 individuals, which are much noisier than integration of a joint distribution of allele age and frequency across a sample of thousands of haplotypes.

## Acknowledgements

This work was funded by NIH grants GM116381 and HG008140, and by the Howard Hughes Medical Institute. We thank Jeffrey Spence and Yun S Song for technical assistance. We also thank Ziyue Gao, Arbel Harpak, Molly Przeworski, Joshua Schraiber, and Aylwyn Scally for comments and discussion, as well as two anonymous reviewers.

## Additional information

### Funding

| Funder | Grant reference number | Author |
| --- | --- | --- |
| National Institutes of Health | NRSA-F32 Grant GM116381 | Kelley Harris |
| Howard Hughes Medical Institute | Investigator Grant | Jonathan K Pritchard |
| National Institutes of Health | R01 Grant HG008140 | Jonathan K Pritchard |

The funders had no role in study design, data collection and interpretation, or the decision to submit the work for publication.

### Author contributions

KH, Conceptualization, Data curation, Software, Formal analysis, Funding acquisition, Validation, Investigation, Visualization, Methodology, Writing—original draft, Writing—review and editing; JKP, Conceptualization, Funding acquisition, Visualization, Writing—original draft, Writing—review and editing

## Author ORCIDs

Kelley Harris, http://orcid.org/0000-0003-0302-2523

## Additional files

### Major datasets

The following previously published datasets were used:

| Author(s) | Year | Dataset title | Dataset URL | Database, license, and accessibility information |
|---|---|---|---|---|
| 1000 Genomes Project Consortium | 2015 | 1000 Genomes Phase 3 | http://www.international-genome.org/category/phase-3/ | Publicly available at internationalgenome.org |
| Swapan Mallick, David Reich, et al | 2016 | Simons Genome Diversity Project | https://www.simonsfoundation.org/life-sciences/simons-genome-diversity-project-dataset/ | Publicly available from the Simons Foundation. Directions for downloading available here: http://simonsfoundation.s3.amazonaws.com/share/SCDA/datasets/2014_11_12/StepstodownloadtheSGDPdataset_v4.docx |
| Prado-Martinez, Tomas Marques-Bonet, et al | 2013 | Whole genome sequences for a set of 79 great ape individuals. Genome sequencing | https://www.ncbi.nlm.nih.gov/bioproject/PRJNA189439/ | Publicly available at NCBI BioProject, submitted as part of the Great Ape Genome Diversity Project (accession no: PRJNA189439) |
| Monkol Lek, Daniel MacArthur, et al | 2016 | Exome Aggregation Consortium | http://exac.broadinstitute.org | Summary data publicly available for download at http://exac.broadinstitute.org/downloads |
| Prado-Martinez, Tomas Marques-Bonet, et al | 2013 | Great Ape Genome Diversity Project | https://www.ncbi.nlm.nih.gov/sra?term=SRP018689 | Publicly available at NCBI Sequence Read Archive (accession no: SRP018689) |

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
