## [Decision Letter]

Thank you for submitting your article "Rapid evolution of the human mutation spectrum" for consideration by *eLife*. Your article has been favorably evaluated by Detlef Weigel (Senior Editor) and three reviewers, one of whom is a member of our Board of Reviewing Editors. The following individual involved in review of your submission has agreed to reveal his identity: Aylwyn Scally (Reviewer #2).

The reviewers have discussed the reviews with one another and the Reviewing Editor has drafted this decision to help you prepare a revised submission.

Summary:

There are often discrepancies in mutation rates and spectra inferred from short term observations and long term comparisons. One potential explanation is that these rates and spectra are not fixed, but vary in the course of evolution. This paper makes an important contribution, by documenting changes in the mutation spectrum during human evolution. It presents an analysis of population differences in the spectrum of context-dependent single nucleotide polymorphism, focused on humans and great apes. A previous observation about differences in mutation spectrum between human populations is replicated here and a hypothesis about a historical burst of mutation is presented. Many additional, weaker differences are also seen but replicated across data sets, which argues for biological, rather than experimental explanations. Very substantial differences among great apes are described.

Essential revisions:

The original reviews are below. In terms of revision, the only major issue that we wish to see addressed concerns the analysis of the burst hypothesis. Specifically, there is a potential concern about whether certain artefacts could explain the apparent lack of evidence for the population-specific bias among the most recent mutations and questions about differences among populations in the relationship between allele frequency and age. There are also suggestions about how to look at patterns along the genome to hunt for clues as to possible causes.

We have left the full reviews in as we think there are other ideas here that you may wish to pick up on, but pursuing them is not essential for the revision. We look forward to seeing the revision.

*Reviewer #1:*

This paper presents an analysis of population differences in the spectrum of context-dependent single nucleotide polymorphism, primarily focused on humans, but also analysing data from great apes. One of the authors previously reported a substantial difference in one particular type. This is replicated here and augmented with a large number of other, much weaker findings. These are replicated across data sets, which argues for biological, rather than experimental explanations. Very substantial differences among great apes are described. One specific hypothesis, about an apobec mutation is assessed and some – moderately weak – evidence for association is seen.

The analyses presented are basically well done and reasonably compelling that there are repeatable differences in mutational spectra. The obvious – and I think rather important – criticism is that this works moves us no further along in terms of identifying causal factors. The authors argue for a contribution of transient mutator phenotypes. However, it is not clear how plausible this model is – most models of mutator suggest that they tend not to persist in sexually reproducing populations. This could potentially be explored by simulation. The authors argue that non-genetic factors – e.g. environmental exposure – are unlikely to explain the phenomena – though no hard evidence is given, although I agree that the great ape differences are compatible with a substantial genetic component.

The only other substantial comment I have is around the analysis that fits the burst of TCC-to-TTC mutations. My concern is that sequencing data sets will have higher error rates at low frequencies and likely differential discovery based on sequence context due to systematic fluctuation in sequencing depth (true for both UK10k and 1000G). Hence, I have a concern that the most recent TCC->TTC mutations could be being lost/swamped in such a way that leads to an apparent burst, when the process is still active.

*Reviewer #2:*

This is a nice paper on a topic of current interest and relevance in human genetics and genomics. It points to important evidence for recent variability over time in at least some aspects of the human mutation rate, something which until now we have only been able to speculate about, and which has potentially broad implications for human evolutionary genetics.

I have listed below a few thoughts and comments, including some things I think the authors should address. However, I found the paper well written and well presented, and have no major issues to raise.

It would be good to get a sense of the raw numbers involved. What do the relative differences between populations mean in terms of actual numbers of variants? For example, what actual density of additional derived T alleles are there in Europe compared to Africa for the TCC->TTC signal?

Is there an issue with ascertainment bias due to demography, in that the allele frequency spectrum varies between populations due to their differing demographic histories, and this might differentially affect the ascertainment of variants in different spectral classes, depending on their relative abundance?

It would be interesting to know how the structure presented in Figure 1 varies with the age of the variants used to construct it (or, as a proxy, their allele frequency). Presumably one would expect the differences between populations to disappear as one excludes more recent variants, as older ones are more likely to be shared. Is this the case?

I think the procedure used to estimate statistical significance, referred to as 'a forward variable selection procedure' needs a better description and motivation. It's not wholly clear to me how the procedure adopted achieves its goal of minimising the interdependence of the tests for each mutation type. It looks to me like some form of partitioned chi-squared test for comparing multiple proportions, but I don't know this statistical literature well – can you cite a useful reference or else explain how you arrived at it? I'd be happy with just testing for a significant overall difference in spectrum between two populations – is testing for significance of individual components of the spectrum really necessary?

For the differences within the great ape tree, did you also compare with an outgroup such as macaque or gibbon? It wasn't clear how you polarise the C-T rate difference as an increase on the Pongine branch, and the CpG rate difference as an increase in Hominines.

*Reviewer #3:*

This article extends previous knowledge on heterogeneity of mutational spectra between populations. It detailed differences in frequencies of specific mutation types between diverse population groups. Moreover, authors date TCC->TTC mutational pulse for the European population. The arguments supporting the main conclusion of the study seem convincing, although the discussion of possible importance of evolutionary forces other than mutation would be helpful (e.g. interaction between BGC and demographic history). There is a noticeable overlap with the earlier work, and the manuscript would strongly benefit from additional analyses of the observed mutational patterns.

For example, are the relative increases of mutation types uniformly distributed along the genome or enriched in specific genomic locations? Are they associated with epigenomic features or display asymmetry with respect to transcription or replication? Are they dependent on local recombination rate? Any analysis suggesting of a mechanistic hypothesis underlying the observation would strengthen the paper.

In contrast to the main result of the manuscript, I am skeptical about the conjecture related to the APOBEC-induced mutagenesis. It is not statistically sound and based on arbitrary thresholds. Is it possible to compare replication asymmetry of APOBEC-like mutations (TC[A/T]-> [C/G]) between populations with different frequencies of variants associated with breast or bladder cancer?

---

## [Author Response]

*Essential revisions:*

*The original reviews are below. In terms of revision, the only major issue that we wish to see addressed concerns the analysis of the burst hypothesis. Specifically, there is a potential concern about whether certain artefacts could explain the apparent lack of evidence for the population-specific bias among the most recent mutations and questions about differences among populations in the relationship between allele frequency and age. There are also suggestions about how to look at patterns along the genome to hunt for clues as to possible causes.*

We thank the editor and reviewers for these thoughtful comments, which have helped us to improve the substance and presentation of the paper. To address this essential revision point, we welcome the chance to present additional evidence that the burst of TCC→TTC mutations at intermediate frequencies is not well explained by bioinformatic artifacts. We have added a new analysis of the Exome Aggregation Consortium Data (Figure 3—figure supplement 2), a dataset even larger than UK10K that supports the mutation burst hypothesis as well as the datasets we previously analyzed. This evidence is summarized in the following new paragraph of the paper):

“Although low frequency allele calls often contain a higher proportion of base calling errors than higher frequency allele calls do, it is not plausible that base-calling errors could be responsible for the pulse we have described. […] When independently generated datasets of different sizes are projected down to the same sample size, the TCC→TTC pulse spans the same range of allele frequencies in both datasets (Figure 3—figure supplement 1 and Figure 3—figure supplement 2).”

Figure 3 and Figure 3—figure supplement 2 illustrate how well supported the pulse pattern is in all cases.

The point about differences between populations in the relationship between allele frequency and age is addressed in direct response to reviewer 2.

*We have left the full reviews in as we think there are other ideas here that you may wish to pick up on, but pursuing them is not essential for the revision. We look forward to seeing the revision.*

We appreciate this encouraging response, and have incorporated many ideas from these reviews into additional supplementary analyses. As described in more detail below, we have added new supplementary heat map figures that describe the variation of the human mutation spectrum with allele frequency, replication timing, chromatin state, and transcription. We have also attempted to clarify the description of our significance-testing procedure in the Methods section. Finally, we decided to remove the APOBEC analysis given that it is less conclusive than the other sections of the paper and appears to have detracted, in the eyes of the reviewers, from the impact of the paper’s main conclusions.

*Reviewer #1:*

*This paper presents an analysis of population differences in the spectrum of context-dependent single nucleotide polymorphism, primarily focused on humans, but also analysing data from great apes. One of the authors previously reported a substantial difference in one particular type. This is replicated here and augmented with a large number of other, much weaker findings. These are replicated across data sets, which argues for biological, rather than experimental explanations. Very substantial differences among great apes are described. One specific hypothesis, about an apobec mutation is assessed and some – moderately weak – evidence for association is seen.*

*The analyses presented are basically well done and reasonably compelling that there are repeatable differences in mutational spectra. The obvious – and I think rather important – criticism is that this works moves us no further along in terms of identifying causal factors. The authors argue for a contribution of transient mutator phenotypes. However, it is not clear how plausible this model is – most models of mutator suggest that they tend not to persist in sexually reproducing populations. This could potentially be explored by simulation. The authors argue that non-genetic factors – e.g. environmental exposure – are unlikely to explain the phenomena – though no hard evidence is given, although I agree that the great ape differences are compatible with a substantial genetic component.*

It is a fair point that we have not yet nailed down concrete mechanisms that are causing mutation spectrum evolution. However, it is our hope that describing the temporal structure of mutation spectrum change will bring us and others closer achieving this goal in the future. Thanks to the work done in this paper, we now know that the TCC*→*TTC pulse appears to be presently inactive, meaning that it is probably not caused by currently segregating genetic variation that could be mapped via genome-wide association approaches. This implies that the TCC*→*TTC pulse might not be the best signal to chase in search of causal factors, which is an important thing to keep in mind for anyone who wants to tackle the challenging problem of mapping mutators in the future. This paper also shows that there are other mutation spectrum differences between populations that are weaker in magnitude than the TCC*→*TTC pulse, but are nevertheless reproducible and might be better leads to go after in search of the molecular underpinnings of mutation rate variation.

In terms of the plausibility of mutator phenotypes existing in sexual populations, a natural mutator phenotype was recently identified in yeast (see “Mis- match repair incompatibilities in diverse yeast populations” by Bui, et al. Genetics 2017).

*The only other substantial comment I have is around the analysis that fits the burst of TCC-to-TTC mutations. My concern is that sequencing data sets will have higher error rates at low frequencies and likely differential discovery based on sequence context due to systematic fluctuation in sequencing depth (true for both UK10k and 1000G). Hence, I have a concern that the most recent TCC->TTC mutations could be being lost/swamped in such a way that leads to an apparent burst, when the process is still active.*

We believe that the new analyses presented in Figure 3—figure supplement 2 and Figure 3—figure supplement 6 provide good evidence that sequencing errors are not artificially deflating estimates of the TCC→TTC mutation fraction at low frequencies.

*Reviewer #2:*

*This is a nice paper on a topic of current interest and relevance in human genetics and genomics. It points to important evidence for recent variability over time in at least some aspects of the human mutation rate, something which until now we have only been able to speculate about, and which has potentially broad implications for human evolutionary genetics.*

*I have listed below a few thoughts and comments, including some things I think the authors should address. However, I found the paper well written and well presented, and have no major issues to raise.*

*It would be good to get a sense of the raw numbers involved. What do the relative differences between populations mean in terms of actual numbers of variants? For example, what actual density of additional derived T alleles are there in Europe compared to Africa for the TCC->TTC signal?*

The raw numbers of TCC→TTC mutations (as well as mutations in other contexts) are available in the supplementary file total_continent_mut_counts.txt. In particular, there are 270,538 TCC→TTC variants in Africa compared to 187,174 in Europe (where there are many fewer total SNPs).

*Is there an issue with ascertainment bias due to demography, in that the allele frequency spectrum varies between populations due to their differing demographic histories, and this might differentially affect the ascertainment of variants in different spectral classes, depending on their relative abundance?*

With regard to the relationship between allele age and allele frequency, it is a fair point that different populations have different relationships between allele frequency and allele age due to contrasting demographic histories. However, differences in the relationship between allele frequency and allele age cannot produce differences between the mutation spectra of two populations in the scenario where each population has experienced an identical rate and spectrum of mutations throughout recent history. The exception to this would be if selective forces like biased gene conversion caused different populations to retain certain classes of mutations at different rates, which is plausible in principle since selective forces act more weakly in populations of smaller effective size. The observed patterns are not consistent with classical biased gene conversion though – if biased gene conversion were the only force creating differences between population mutation spectra, all C/G⇌A/T mutations would be affected in much the same way regardless of sequence context and neither C⇌G nor A⇌T mutations should have differences in abundance between populations.

*It would be interesting to know how the structure presented in Figure 1 varies with the age of the variants used to construct it (or, as a proxy, their allele frequency). Presumably one would expect the differences between populations to disappear as one excludes more recent variants, as older ones are more likely to be shared. Is this the case?*

This is a nice suggestion for bridging the visualizations presented in Figure 1 and Figure 2. We now include this analysis as Figure 3—figure supplement 6, and it indeed shows that differences disappear as we restrict to higher frequency variants.

*I think the procedure used to estimate statistical significance, referred to as 'a forward variable selection procedure' needs a better description and motivation. It's not wholly clear to me how the procedure adopted achieves its goal of minimising the interdependence of the tests for each mutation type. It looks to me like some form of partitioned chi-squared test for comparing multiple proportions, but I don't know this statistical literature well – can you cite a useful reference or else explain how you arrived at it? I'd be happy with just testing for a significant overall difference in spectrum between two populations – is testing for significance of individual components of the spectrum really necessary?*

Since this procedure is not described in the literature to our knowledge, we have expanded its description in the supporting information to include better motivation for what is done (see Methods section). These p-values are used to annotate Figure 1 with the dots that denote significance. We think that conservatively estimating the number of mutation types that vary in rate between populations is important to this paper because Harris 2015 focused on the rate distribution of a single mutation type and we want to emphasize here that mutation spectrum evolution is much more pervasive than that.

*For the differences within the great ape tree, did you also compare with an outgroup such as macaque or gibbon? It wasn't clear how you polarise the C-T rate difference as an increase on the Pongine branch, and the CpG rate difference as an increase in Hominines.*

We have rephrased the statement about CpGs to say, “one major trend is a higher proportion of CpG mutations among the species closest to human” to be more agnostic about the question of whether the rate decreased in one part of the tree versus increased in another part of the tree. For statements about A→T and A→C mutations in the great apes, we have left in language that hypothesizes increases and decreases in rate because these are supported by parsimony (i.e., the rate differential is more parsimoniously explained by a rate increase along one branch of the tree than by rate decreases along two separate branches).

*Reviewer #3:*

*This article extends previous knowledge on heterogeneity of mutational spectra between populations. It detailed differences in frequencies of specific mutation types between diverse population groups. Moreover, authors date TCC->TTC mutational pulse for the European population. The arguments supporting the main conclusion of the study seem convincing, although the discussion of possible importance of evolutionary forces other than mutation would be helpful (e.g. interaction between BGC and demographic history). There is a noticeable overlap with the earlier work, and the manuscript would strongly benefit from additional analyses of the observed mutational patterns.*

*For example, are the relative increases of mutation types uniformly distributed along the genome or enriched in specific genomic locations? Are they associated with epigenomic features or display asymmetry with respect to transcription or replication? Are they dependent on local recombination rate? Any analysis suggesting of a mechanistic hypothesis underlying the observation would strengthen the paper.*

We agree and, we have included new supplementary figures showing how these patterns vary with replication timing, chromatin state, and transcriptional state. Our main inference from these figures is that the discernible features of Figure 1 appear to vary only modestly across the genome.

*In contrast to the main result of the manuscript, I am skeptical about the conjecture related to the APOBEC-induced mutagenesis. It is not statistically sound and based on arbitrary thresholds. Is it possible to compare replication asymmetry of APOBEC-like mutations (TC[A/T]-> [C/G]) between populations with different frequencies of variants associated with breast or bladder cancer?*

We agree that these conclusions about APOBEC-induced mutagenesis are not as decisive as the main conclusions in the paper. We intended this analysis to work as a useful illustration of how linkage disequilibrium can be used to test hypotheses about mutator activity, but in revising the paper, we have decided to remove this section to avoid detracting from the impact of the more key points we are making.